# Deep Learning-Based Feature Extraction of Acoustic Emission Signals for Monitoring Wear of Grinding Wheels

**DOI:** 10.3390/s22186911

**Published:** 2022-09-13

**Authors:** D. González, J. Alvarez, J. A. Sánchez, L. Godino, I. Pombo

**Affiliations:** 1Department of Mechanical Engineering, Faculty of Engineering of Bilbao, University of the Basque Country (UPV/EHU), 48013 Bilbao, Spain; 2Ideko Centro Tecnológico, Basque Research and Technology Alliance (BRTA), 20870 Elgoibar, Spain

**Keywords:** deep learning, acoustic emission, grinding, feature extraction

## Abstract

Tool wear monitoring is a critical issue in advanced manufacturing systems. In the search for sensing devices that can provide information about the grinding process, Acoustic Emission (AE) appears to be a promising technology. The present paper presents a novel deep learning-based proposal for grinding wheel wear status monitoring using an AE sensor. The most relevant finding is the possibility of efficient feature extraction form frequency plots using CNNs. Feature extraction from FFT plots requires sound domain-expert knowledge, and thus we present a new approach to automated feature extraction using a pre-trained CNN. Using the features extracted for different industrial grinding conditions, t-SNE and PCA clustering algorithms were tested for wheel wear state identification. Results are compared for different industrial grinding conditions. The initial state of the wheel, resulting from the dressing operation, is clearly identified for all the experiments carried out. This is a very important finding, since dressing strongly affects operation performance. When grinding parameters produce acute wear of the wheel, the algorithms show very good clustering performance using the features extracted by the CNN. Performance of both t-SNE and PCA was very much the same, thus confirming the excellent efficiency of the pre-trained CNN for automated feature extraction from FFT plots.

## 1. Introduction

Tool wear monitoring is a critical issue in advanced manufacturing systems [1]. High-end machining equipment for turning, milling and other material removal processes require the development of new technologies that allow efficient monitoring of tool wear [2]. However, existing industrial wear monitoring methods are somewhat stationary, given that these are designed on the basis that cutting conditions do not vary over a given time interval. Consequently, new methods are being proposed to overcome this limitation [3]. With the advent of Cyber-Physical Systems (CPS) in the manufacturing sector, real-time tool wear monitoring has become essential for achieving efficient system integration [4]. Due to the complexity of the problem and the variability between existing tool wear patterns, the full exploitation of artificial intelligence (AI)-based approaches is a new and powerful tool for wear monitoring [5,6]. The final objective is to avoid machine halt due to the need for off-line tool inspection and the use of excessively conservative machining parameters that reduce the performance of the complete manufacturing chain.

Of the available machining technologies, grinding plays a critical role in high-tech manufacturing companies [7]. Examples of high-added value products that require grinding in their manufacturing chain include transmission elements for the car industry [8], finishing of landing gear, NGVs and turbine blades for the aerospace sector [9], and difficult-to-machine components for shipbuilding shops [10]. The application window of grinding is growing day after day, to the point where material removal rates comparable to those of milling in extremely hard part materials are achieved [11]. At the same time, the requirements for surface quality (including not only a very smooth surface finish, but also premium surface integrity) and tight tolerances in many mechanical elements can only be met by advanced grinding processes [12].

The wear of the grinding wheel determines the performance of the operation [13,14]. Due to extreme local pressures (1–2 GPa) in the contact between abrasive grits and part material, and high temperatures, the grinding wheel degrades. Various wear micro-mechanisms that largely affect wheel performance may be present, namely wear by attrition, grain pull-out, and grain breakage. Debris may also occupy pores in the grinding wheel, leading to wheel loading. Whatever the case, the grinding operation is affected by these phenomena, and dressing is required to recover the original topography of the grinding wheel. Wear monitoring allows effective decision-making on the need for dressing, and results in higher productivity and optimization of the service life of the grinding wheel. In industry, most of the time the dressing interval is set on the basis of indirect observations such as part surface finish, part tolerances, or even the occurrence of grinding burn. This is a conservative approach that operates far before the point at which the grinding wheel has reached its maximum service life. A complete review of direct and indirect (including accelerometers, force sensors, and acoustic emission) monitoring methods in grinding can be found in [15].

Acoustic emission (AE) is a non-destructive testing technique that has been used since 1950 [16]. Industrial state-of-the-art grinding machines incorporate this technology mainly for contact detection, but in recent decades extensive research work has been dedicated to using AE for the effective on-line characterization of wheel wear. AE sensors are superior to accelerometers in the higher sensitivity and sampling frequency range [17]. On the other hand, because of these characteristics, AE records can be severely affected by multiple existing phenomena at the contact zone (e.g., the coolant jet) generating a signal with high levels of background noise [18]. For these reasons, significant research efforts are currently focused on the problem of correct representation of the original AE signal [19]. The use of machine learning (ML) algorithms for automated feature extraction is a promising alternative that deserves analysis. In other related fields, such as synthetic speech detection, the use of the SincNet architecture is used [20]. SincNet has a number of advantages over standard convolutional networks, namely, fewer parameters, fewer operations to calculate the filter fast convergence and good interpretability of the network.

The objective of feature extraction is to translate raw data into a more compact form while maintaining the most relevant components of the original information. In general, ML algorithms have proven to be more effective when using selected representative features rather than raw data. While manual feature extraction has been carried out for decades by experts in this field, in recent years (and mainly for image applications) some layers of deep learning (DL) networks have been successfully used to automatically extract the most relevant features from data [21].

One of the first attempts to automate feature extraction from AE and other sensors using ML algorithms can be found in [22]. This study used a feed-forward backpropagation neural network for feature selection, yielding very good values (higher than 80%) of accuracy in classification of wheel wear state. Because of the characteristics of the AE waveform, frequency domain analysis has notable limitations [23]. Time-frequency analysis has instead been shown to be robust in the case of AE signals. In the field of time-frequency analysis, the Short-Time Fourier Transform (STFT) and the Wavelet Transform (WT) are powerful tools for signal analysis [24]. WT uses wavelets to build a frequency spectrum as a function of time, thus containing more information than the power-density spectrum obtained from the Fourier series. Further, wavelet analysis is superior to STFT in those multiple resolutions in which it is possible to measure frequency and time scales. In the first decade of the 2000s, the earliest studies were published on grinding wheel wear identification using WT analysis of AE waveforms [25,26]. Wavelet-based feature extraction of the AE signal was used to feed supervised ML models, yielding highly accurate results in controlled laboratory experiments. In [27], discrete wavelet transform was proposed for wear monitoring with excellent results in terms of classification accuracy. In [28], manually extracted features of various sensors, including AE, were used to feed different neural network architectures, again with good classification results. Nonetheless, further research work is required to fully automate feature extraction from the AE signal.

Very recent research work has demonstrated the feasibility of using the information contained in 2-D images generated by the Continuous Wavelet Transform (CWT) [29]. The authors used a Deep ResNet architecture to automatically extract relevant features and then used the data as input for an Extreme Learning Machine (ELM). In other fields such as medicine [30], wavelet scattering using scalograms generated by CWT has shown great potential for automatic feature extraction and subsequent application of supervised ML algorithms. Supervised approaches require extensive labelling of data, which is difficult and time-consuming in the case of wear of grinding wheels. To the best knowledge of the authors, unsupervised clustering of wear status of the grinding wheel has not yet been carried out. Unsupervised clustering avoids the need for data labelling and reduces the computational cost of ML solutions, and therefore opens a new and interesting perspective to the problem of wear monitoring of grinding wheels.

This paper presents a novel machine learning-based proposal for the monitoring of grinding wheel wear status monitoring. The AE technique, which is a robust and well-known sensor in the grinding industry, is used because this sensor is already present in many state-of-the-art industrial grinding machines. The information of the AE signal is compacted in the frequency domain. The relevant features of the Fourier transform images are then extracted using an existing pre-trained deep neural network. Arbitrariness and domain expert in feature extraction are thus efficiently avoided. Subsequently, Principal Component Analysis (PCA) and t-Distributed Stochastic Neighbor Embedding (t-SNE) techniques are combined to produce the clustering algorithm. The results are compared for different industrial grinding conditions. Section 2 presents the experimental methodology and the grinding experiments, while Section 3 deals with data generation, the deep learning based feature extraction strategy, and the proposal for clustering algorithms. Finally, the discussion of results is presented in Section 4. The initial state of the wheel resulting from the dressing operation is clearly identified for all the experiments carried out. This is a very important result since dressing strongly affects operation performance. When grinding parameters produce acute wear of the wheel, the algorithms exhibit very good clustering performance using the features extracted by the CNN. Performance of both t-SNE and PCA is very much the same, thus confirming the excellent efficiency of the pre-trained CNN for automated feature extraction from FFT plots

## 2. Experimental Methodology

Surface grinding experiments were conducted on a three-axis CNC surface grinding machine Blohm Orbit 36. The coolant was 5% water-based oil emulsion, and the part material was ASTM A681 02 (DIN 90MnCrV8) tool steel hardened at 54HRc. The grinding wheel was vitrified with 100% cubic monocrystalline alumina (as for ∅375 × 127 × 40 60H12V489P24P, UNESA S.L., Hernani, Spain). The conditions for the grinding experiments are displayed in Table 1. The values were selected considering actual industrial application of the grinding wheel.

The objective of the experiments was to determine if the AE signal contains valuable information about the wear state of the grinding wheel throughout its service life. Therefore, for each test, grinding passes were repeated until wear reached a certain point at which the experiment needed to be stopped (for instance, the occurrence of vibrations). Dressing was only performed at the beginning of each experiment to ensure a brand-new wheel topography that was similar across the three experiments. Acoustic emission signals were measured using a Nordmann Sea-Mini sensor, which is a very suitable measuring element to collect acoustic signals in machining processes, such as milling, turning, and grinding operations. This is because its strong point is the detection of contact between the part and the cutting tool. Its working frequency range is between 5 kHz and 1 MHz, its dynamic amplitude does not exceed 110 dB, and its optimal operating range is between −40 and 85 °C. The sensor can only be used with its corresponding acquisition, processor, and power supply cards since, among other things, it must be connected to a source of no more than +/−15 V. The sensor is fixed to the magnetic table as close as possible to the contact region between wheel and part material (see Figure 1). Then, the signals are received by a data acquisition card (NI 6366) with 8 channels and a time resolution of 10 ns, so that the AE values can be processed with a computer.

Figure 2 shows an example of the AE signal collected during Test No. 1. A sample rate of 1 MHz was set for the sensor. Since the duration of one grinding pass was 0.28–0.5 s (depending on the federate of the test), as many as 500,000 data (in the worst-case scenario) were gathered for each grinding pass.

During the tests, as well as the AE signal, other variables that provide indirect information about the wear state of the wheel were also measured, namely part surface roughness and spindle power. Surface roughness was measured using a Surtronic Duo device. Power signals were collected using a Load Control model UPC-230, connected with a data acquisition card (National Instruments NI 6009-USB). Data were analyzed using a program developed in Python.

Analysis of the experimental results revealed different wear patterns for the three grinding tests. Figure 3 shows the evolution of the specific grinding energy *e_c_* as a function of the volume of part material removed (V’_w_). This variable is directly related to the number of grinding passes. The value of e_c_ (J/mm^3^) is a measure of the energy required to remove the unit volume of part material, and provides indirect evidence of the wear state of the grinding tool. Increased values of *e_c_* are related to the loss of removal ability of the wheel due to wear. Calculation of *e_c_* was accomplished by using Equation (1), using as inputs the power measured during the process (*P*) and the grinding parameters (b_s_, a_e_, v_w_):(1)$$e_{c}=\frac{P}{b_{s}\cdot a_{e}\cdot v_{w}}$$

Figure 3 displays relevant information about wheel performance and the influence of wheel wear under different grinding conditions. The first observation of note is that using the conditions of Test No.2, the specific volume of part material removed, V’w, is considerably greater than that in Tests No.1 and 3. Under the conditions of Test No.1 and 3, wheel wear vibrations appeared when V’_w_ was equal to 628.42 mm^3^/mm (Test No.1) and 450.50 mm^3^/mm (Test No.3), and therefore the tests had to be stopped. Instead, for Test No.2, vibrations only appeared for a larger volume of part material removed (V’_w_ = 1228.85 mm^3^/mm). Concerning the evolution of e_c_, there was a clear upward trend on Test No.2, with the values increasing from an initial figure of 61 J/mm^3^ up to a maximum of 171 J/mm^3^ (maximum difference 90 J/mm^3^) thus showing a clear effect of wheel wear on process performance. However, for Tests No.1 and 3, the values of e_c_ showed a more constant trend, the maximum variation being 33 J/mm^3^ for Test No. 1 and 27 J/mm^3^ for Test No.3.

Results of surface roughness provided additional experimental evidence regarding the influence of wheel wear on process performance. Figure 4 plots part surface roughness (R_a_, μm) as a function of the specific volume of part material removed, V’_w_, for the different tests. As stated above, the behavior during Tests No.1 and 3 was conditioned by the occurrence of vibrations that signal the end of the experiments. In both cases, a rapid increase in surface roughness was evident. In the case of Test No.2, the slope of the curve was less steep, but continuous throughout the experiment. Because of the variations in the length of the tests, it can be concluded that in the case of Tests No.1 and 3 the influence of wheel dressing is more acute, whereas in the case of Test No.2, together with initial wheel dressing, wheel wear also plays a relevant role.

In view of the above results, a strategy was proposed that involved clustering for classification of the wear stage of the grinding wheel. Thus, three regions were identified related to the wear state of the grinding wheel, namely initial stage, intermediate stage and end of service life. Because the lengths of the tests were different, the extension of each region also needed to be defined:
-Test No.1: initial stage from V’_w_ = 0 to V’_w_ = 110.37 mm^3^/mm (25 AE records); intermediate stage from V’_w_ = 257.91 mm^3^/mm to V’_w_ = 330.81 mm^3^/mm (27 AE records); end of service life from V’_w_ = 481.16 mm^3^/mm to V’_w_ = 628.42 mm^3^/mm (29 AE records).-Test No.2: initial stage from V’_w_ = 0 to V’_w_ = 185.21 mm^3^/mm (50 AE records); intermediate stage from V’_w_ = 484.33 mm^3^/mm to V’_w_ = 631.87 mm^3^/mm (50 AE records); end of service life from V’_w_ = 1042.37 mm^3^/mm to V’_w_ = 1228.85 mm^3^/mm (50 AE records).-Test No.3: initial stage from V’_w_ = 0 to V’_w_ = 150.83 mm^3^/mm (40 AE records); intermediate stage from V’_w_ = 185.93 mm^3^/mm to V’_w_ = 299.71 mm^3^/mm (39 AE records); end of service life from V’_w_ = 337.18 mm^3^/mm to V’_w_ = 450.50 mm^3^/mm (40 AE records).

It is important to note that the definition of these regions is arbitrary, which means there is a possibility of observing no difference between some records belonging to adjacent stages. The objective was to test whether the clustering strategy can detect and classify the differences; the degree of arbitrariness is addressed in the subsequent discussion.

## 3. Data Generation, DL-Based Feature Extraction, and Clustering Algorithms

### 3.1. Data Generation

The literature review of Section 1 highlights the difficulties in extracting valuable information from AE signals, and this limitation is even more obvious for grinding operations. In fact, the use of AE sensors in state-of-the-art industrial grinding machines is mostly focused on contact detection, where the AE sensor exhibits very high reliability.

Several authors have proposed the use of time-domain analysis (for instance, the Wavelet Transform) to extract information from AE sensors, especially for event detection in a given time domain. However, for the problem proposed in this work, the WT is not a good choice. The reason is that during a single grinding pass there is not enough accumulated wear to reveal differences in the scalogram of the signal. For this reason, we expected to find a very constant scalogram plot, which might account for the transient phenomena at the beginning and the end of the grinding pass. These events are not related to the wear stage of the grinding wheel, and therefore, such information is not useful for solving the proposed clustering strategy. A good example can be found in Figure 5, where the scalogram of a grinding pass during Test No. 2 is plotted. Data analysis was carried out using MATLAB^®^. The energy accumulated in the different frequencies was constant except for the yellow flashes at the beginning and end of the pass. This result also confirms the capacity of the AE sensor to detect contact.

We decided to work in the frequency domain using the Fast Fourier Transform (FFT). The hypothesis is that the frequency spectrum must account for the differences in the wheel topography due to wear throughout the passage of time. In other words, the frequency spectra in the *initial stage*, the *intermediate stage* and the *end of service life* must contain different information. However, this is not so obvious upon inspection of those spectra, which contain information about a large number of different events. Following the results of Griffin and Chen [31], who studied the AE signal in single-grain experiments, it is known that the phenomena related to the interaction grain-machined surface can be found within the frequency range 50–500 KHz. Therefore, it was decided to filter the signal and discard the information outside the 50–500 KHz interval. Moreover, from the 500,000 data corresponding to the complete grinding pass, only 300,000 corresponding to the center part of the pass were selected, avoiding the entrance and the end of the pass. Figure 6 represents the FFT of the AE signal of one grinding pass during Test No. 2.

The information of the AE signal is compacted in the frequency domain in the form of FFT plots. The relevant features of the Fourier transform images are then extracted using ResNet 18, an existing pre-trained deep neural network. Arbitrariness and domain expert in feature extraction are thus efficiently avoided. Subsequently, Principal Component Analysis (PCA) and t-Distributed Stochastic Neighbor Embedding (t-SNE) techniques were combined to produce the clustering algorithm.

Following the above procedure, FFTs for all the grinding passes at the different wear stages (as defined above) were obtained, that is, 81 plots for Test No. 1, 150 for Test No. 2, and 150 for Test No. 3. Each figure was then normalized so that size, color, and limits of the axis were identical. The aim was to limit the differences in the information contained within the frequency spectra.

### 3.2. Deep Learning-Based Feature Extraction

ResNet-18—a pretrained Convolutional Neural Network (CNN) from MATLAB^®^—was used to extract features from the frequency plots. A CNN is a typical supervised feed-forward deep learning network. The architecture is made up of different types of layers, including convolutional, pooling and fully-connected layers. Convolutional and pooling layers are key for feature extraction and for preventing overfitting. For this reason, CNNs are commonly used for image clustering, with excellent results.

However, CNNs are prone to the problems of gradient vanishing and can even explode during training, which strongly affects their performance. In order to solve these problems, He et al. proposed the concept of Residual Network or Residual Learning (RL) [32], which is based on the technique of skip connection (skipping some of the network layers to obtain a direct connection to the output). ResNet-18 is a good example of a high-performance RL network, available after having been trained using more than one million images. Because of the large number of different images used (such as many objects and animals), it can provide solid feature representations of various types of images. ResNet-18 is 18 layers deep and the picture input size is 224 × 224 [33].

ResNet18, which is known for its very good performance in image recognition, was fed with the plots of the FFTs. Thus, the input for ResNet18 was a 2D value of dimension 224*224.The output from ResNet18 were the features extracted from the 2D FFT plots. Features were stored in a 2D value whose dimension depends on the number of samples of each test. Thus:
-Test 1: dimension of Feature 2D value 1000 × 162-Test 2: dimension of Feature 2D value 1000 × 300-Test 3: dimension of Feature 2D value 1000 × 162

The above Feature 2D values were the inputs for both PCA and t-SNE for dimension reduction.

### 3.3. Clustering Algorithms

After feature extraction, clustering algorithms were used to classify the different wheel wear stages. Two methods were proposed in this study: t-distributed Stochastic Neighbor Embedding (t-SNE) and Principal Component Analysis (PCA). t-SNE is a statistical algorithm that is highly efficient for visualizing high-dimension data [34]. The method assigns a location on a 2D or 3D map to each dataset. In fact, it is a variant of the Stochastic Neighbor Embedding method proposed in [35]. t-SNE has proven its potential in many application fields, with strong implantation in signal processing.

PCA is another powerful tool for dimensionality reduction. The method is based on the projection of each dataset on the first principal components. Therefore, the algorithm works very well when many of the variables exhibit large correlations, and the objective is to reduce their number to an independent set.

## 4. Discussion of Results

Unsupervised clustering of the frequency plots resulting from the various wear stages of Tests No.1, 2 and 3 was carried out using the techniques introduced in the previous section, namely t-SNE and PCA.

Figure 7 graphically displays the t-SNE analysis for Test No.1. Green dots correspond to the *initial stage*, blue dots to the *intermediate stage*, and red dots to the *end of service life*. The results indicate that the *initial stage*, which is the most clearly affected by the initial dressing process, can be readily separated from the other two stages. This means that the clustering algorithm can adequately identify the features related to the effect of dressing on the grinding wheel. However, although a certain trend can be observed, some red dots appear to be mixed with the blue dots. In other words, there is a certain degree of intersection between the intermediate stage and the end of service. This is a logical result, considering the conclusions of Section 2, in which it was shown that for Test No.1, the effect of dressing was more pronounced than that of wheel wear throughout the test.

Figure 8 plots the PCA analysis for Test No. 1. Again, the algorithm is very efficient in clustering the green dots, which correspond to the performance of the wheel just after dressing. The failure to achieve effective separation, particularly between blue dots (*intermediate wear stage)* and red dots (*end of service life*) is again evident. In fact, a number of cases (in the upper region of Figure 8) exhibit strong independence with respect to the rest of the results. Closer inspection of the results reveals that those cases correspond exactly to the first nine passes immediately after dressing. This result is compatible with the performance of the t-SNE algorithm shown in Figure 7.

Clustering results for Test No. 2 are represented in Figure 9 (t-SNE) and Figure 10 (PCA analysis). Again, the t-SNE algorithm provides a very efficient identification of the passes corresponding to the *initial stage* (green dots), which can be clearly separated from the passes of the *intermediate stage* (blue dots). Separation between the *intermediate stage* and the *end of service life* is also efficient, although a number of cases are still misclassified. Together with the graphic representation of the results, a criterion was defined to establish the risk of misclassification. A level of high-risk is assigned to those cases which appear mixed with non-adjacent stages, such as, for instance, when cases of the *initial stage* cannot be separated from cases of *end of service life*. Instead, those cases of adjacent stages that cannot be separated lie in the level of medium-risk. Figure 9 shows that none of the misclassifications are of high-risk level. Even for the cases of the *intermediate stage* and the *end of service life*, good clustering behavior is established. The number of medium-risk misclassifications is 1 for the *intermediate stage* and 10 for the *end of service life*. These results are in accordance with the observations of Section 2, which show the greater influence of wheel wear on this test when compared to Tests No.1 and 3.

The PCA analysis for Test No.2 confirms these results, although it should be noted that the clustering performance of the method is not as efficient as that of the t-SNE algorithm for this case. Again, the green dots corresponding to the passes of the *initial stage* appear clearly separated, although 5 cases cannot be separated from the blue dots of the *intermediate stage*. As the experiment progresses, this tendency toward efficient clustering is still observed, although separation between the *intermediate stage* and the *end of service life* is less clear than that provided by t-SNE. There is a region where the algorithm cannot clearly separate the wear stages, although it is worth noting that no case shows a high-risk of misclassification.

The clustering results for Test No.3 are presented in Figure 11 and Figure 12. Both t-SNE and PCA confirmed their excellent capacity to detect the influence of dressing on grinding performance. For both algorithms, the green dots appear clearly separated from blue and red dots. As stated previously, in Test No. 3 the influence of wheel wear was less evident, and thus the misclassification was concentrated in the *intermediate stage* and the *end of service life*. Although this is a medium-risk error, it became clear that the algorithms could not efficiently separate the passes corresponding to this part of the test.

Results show that the method was efficient at separating the signals related to the initial stage. The physical meaning of this result was related to the fact that the initial stage was clearly conditioned by the dressing process of the grinding wheel. Dressing modifies the wheel topography and the active number of grains with respect to the standard performance of the grinding wheel. When the effect of dressing disappears, after a certain number of grinding passes (which cannot be known a priori), the only changes on the wheel surface are related to wear, which does not have as strong an effect as dressing. That is why separating the intermediate stage from the end of service life is more difficult.

Although further experiments will be carried out, existing results suggest that using this method the extension of influence of the dressing process can be effectively detected no matter the grinding conditions. Since the AE signal depends mainly on the contact wheel-workpiece (as shown in the Introduction, the phenomena of rubbing and ploughing have been clearly identified using AE by Chen), and that contact strongly depends on wheel topography and only to a lesser extent, on grinding conditions, it can be expected that the method is suitable for other grinding conditions.

## 5. Conclusions

This article presents a novel ML-based proposal for monitoring grinding wheel wear status using an AE sensor. The most notable finding is the possibility of efficient feature extraction form frequency plots using CNNs. First, it was shown that for the application of surface grinding, in which no difference in wheel wear occurs during one grinding pass, FFT plots can be used instead of CWT scalograms. Feature extraction from FFT plots requires sound domain-expert knowledge, and thus a new approach of automated feature extraction using a pre-trained CNN was presented. Using the features extracted for various industrial grinding conditions, t-SNE and PCA clustering algorithms were tested for wheel wear state identification. Results have been discussed for experiments under three different industrial grinding conditions. The initial state of the wheel, resulting from the dressing operation, was clearly identified for all the experiments carried out. This is a very important result since dressing strongly affects operation performance. When grinding parameters produce clear wear of the wheel (for instance, in test No.2), the algorithms exhibited a very good clustering performance using the features extracted by the CNN. The t-SNE and PCA show very similar performance, thus confirming the excellent efficiency of the pre-trained CNN for automated feature extraction from FFT plots. As a future line of research, the possibility of 3D clustering and other more efficient clustering algorithms to improve the results (such as Uniform Manifold Approximation and Projection for Dimension Reduction, UMAP) is proposed.

## Figures and Tables

**Figure 1 sensors-22-06911-f001:**
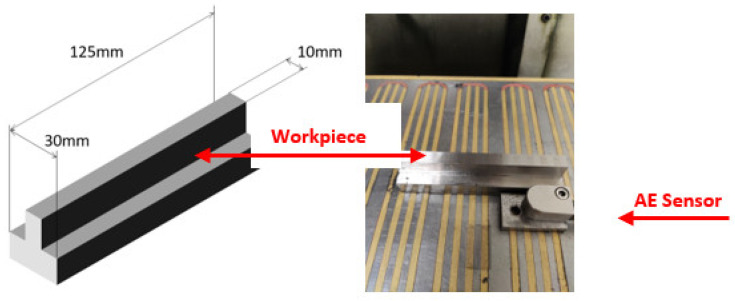
(**Left**): Steel part geometry. (**Right**): Set-up of the AE sensor on the magnetic table.

**Figure 2 sensors-22-06911-f002:**
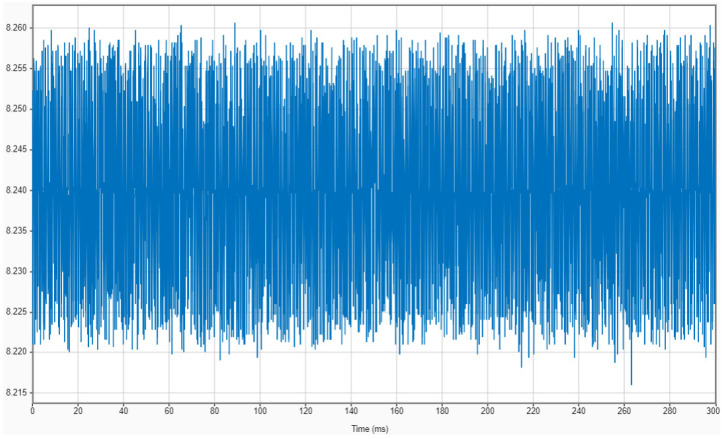
AE signal vs. time during one grinding pass (Test No. 1).

**Figure 3 sensors-22-06911-f003:**
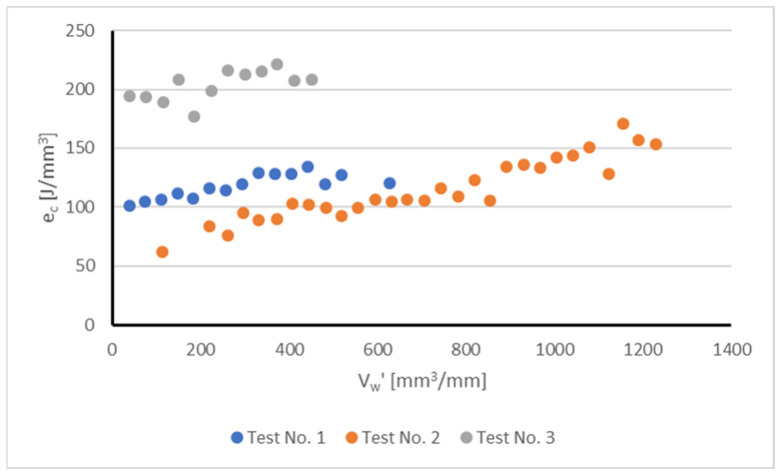
Experimentally measured specific grinding energy e_c_ (J/mm^3^) as a function of the specific volume of part material removed V’_w_ (mm^3^/mm) during the tests.

**Figure 4 sensors-22-06911-f004:**
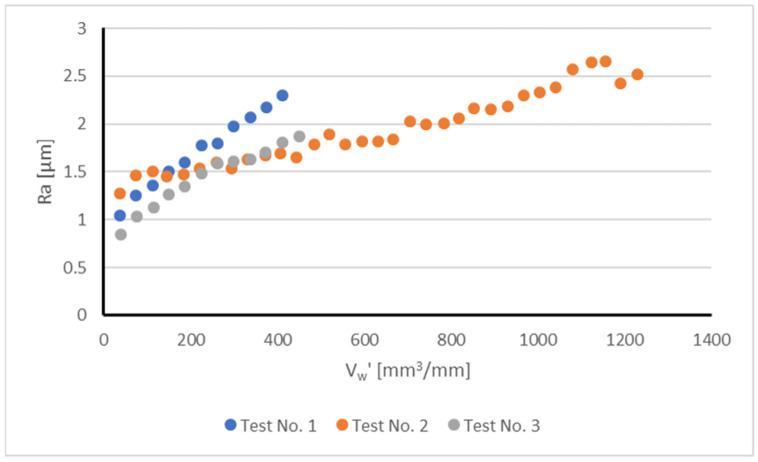
Average surface roughness R_a_ (μm) as a function of the specific volume of part material removed V’_w_ (mm^3^/mm) during the tests.

**Figure 5 sensors-22-06911-f005:**
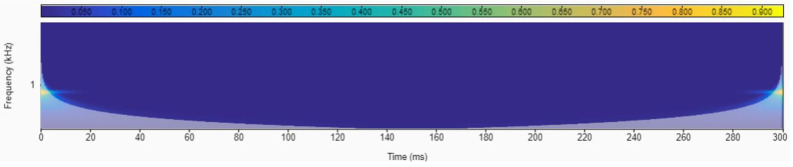
Scalogram of the AE signal during one grinding pass (Test No.2).

**Figure 6 sensors-22-06911-f006:**
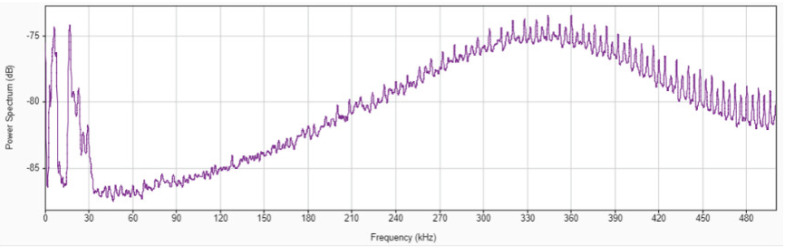
FFT of the AE signal during one grinding pass during Test No.2.

**Figure 7 sensors-22-06911-f007:**
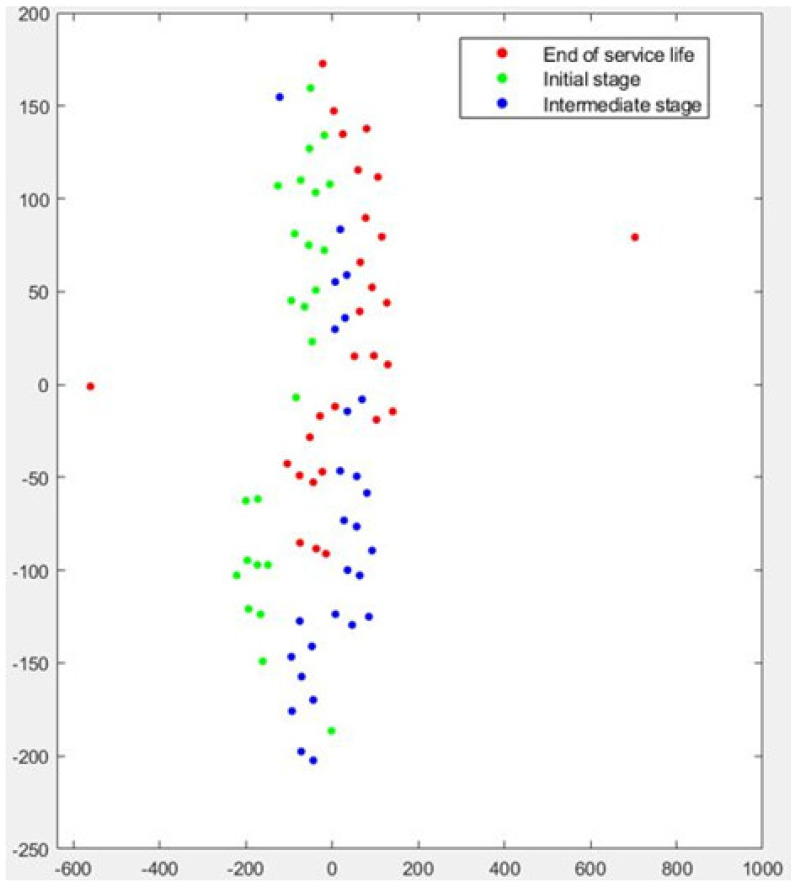
Graphical results of t-SNE dimensionality reduction for Test No. 1.

**Figure 8 sensors-22-06911-f008:**
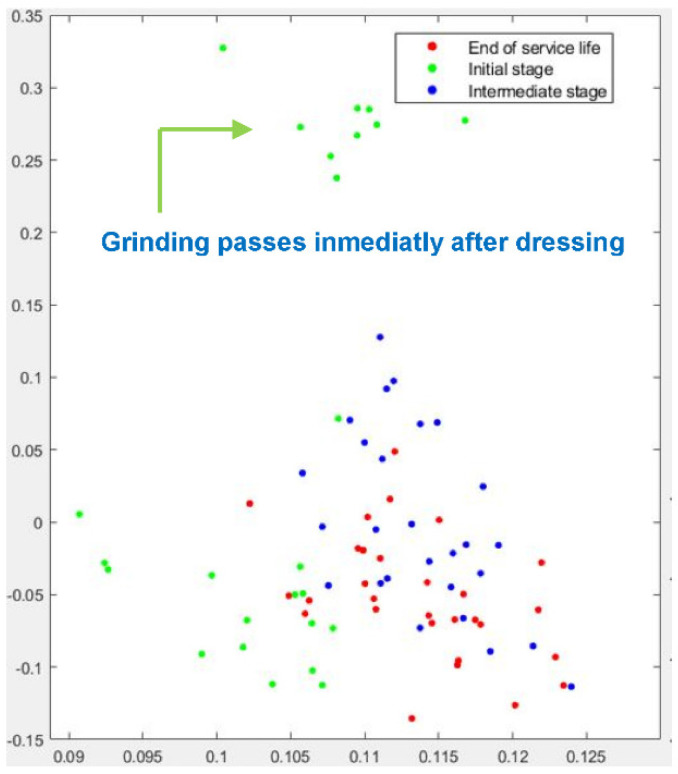
Graphical results of PCA analysis for Test No. 1.

**Figure 9 sensors-22-06911-f009:**
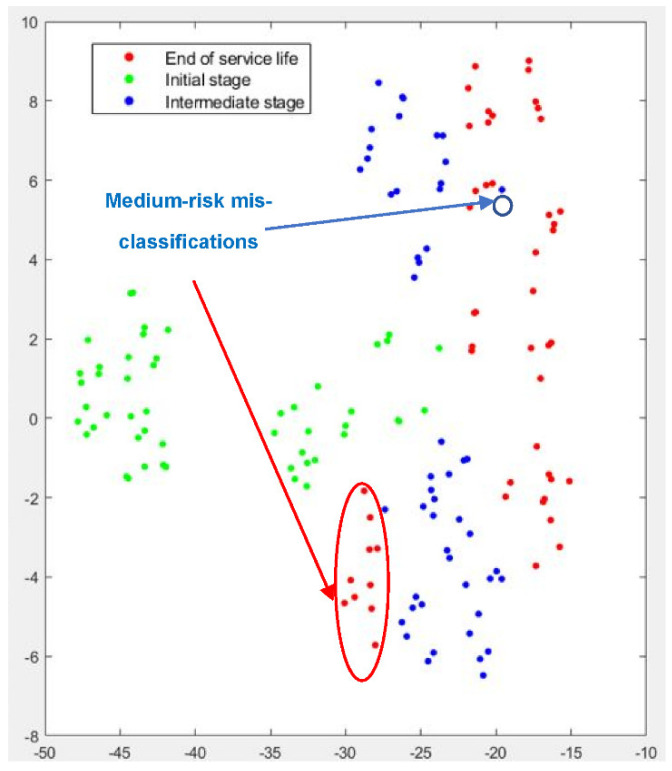
Graphical results of t-SNE dimensionality reduction for Test No. 2.

**Figure 10 sensors-22-06911-f010:**
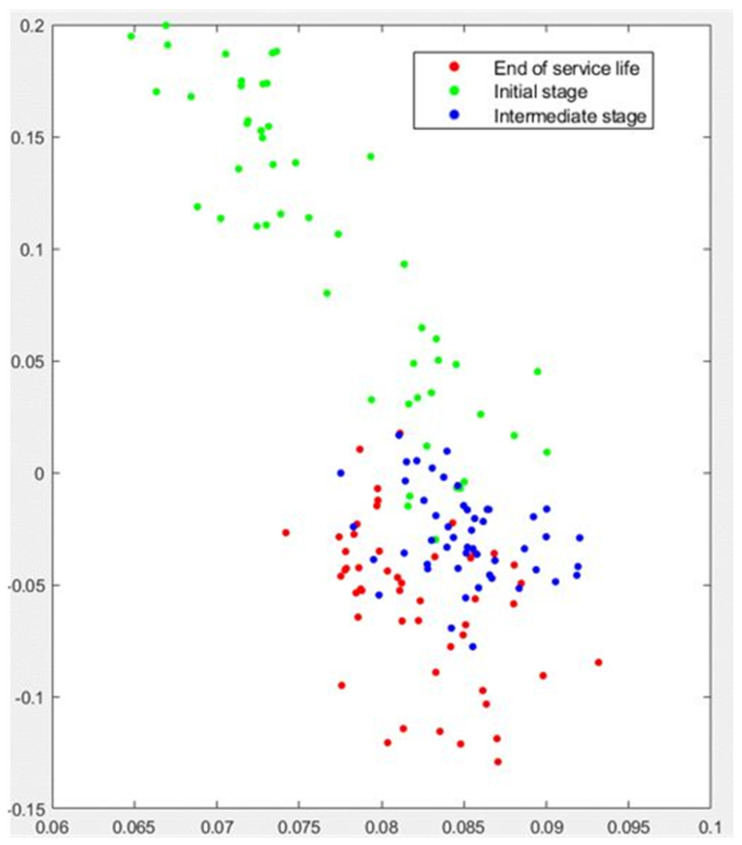
Graphical results of PCA analysis for Test No. 2.

**Figure 11 sensors-22-06911-f011:**
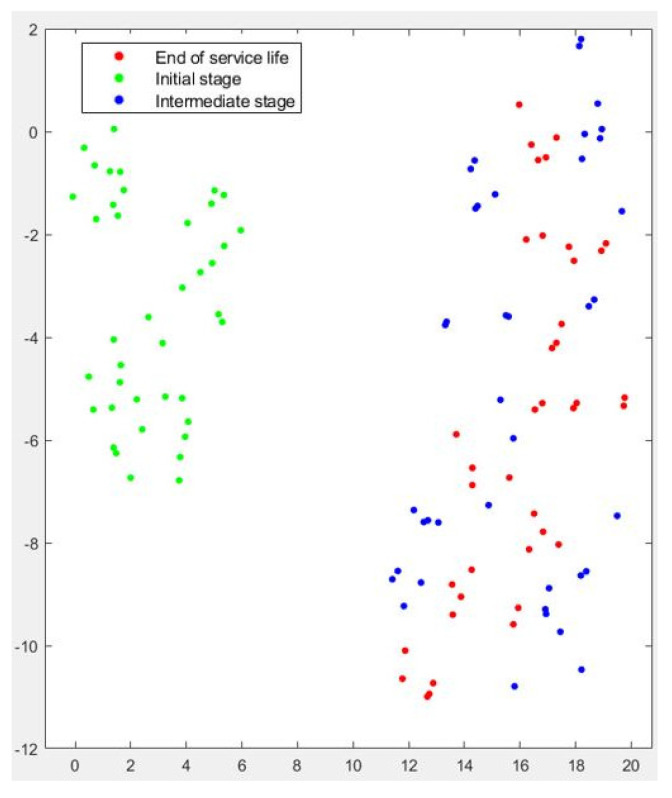
Graphical results of t-SNE dimensionality reduction for Test No.3.

**Figure 12 sensors-22-06911-f012:**
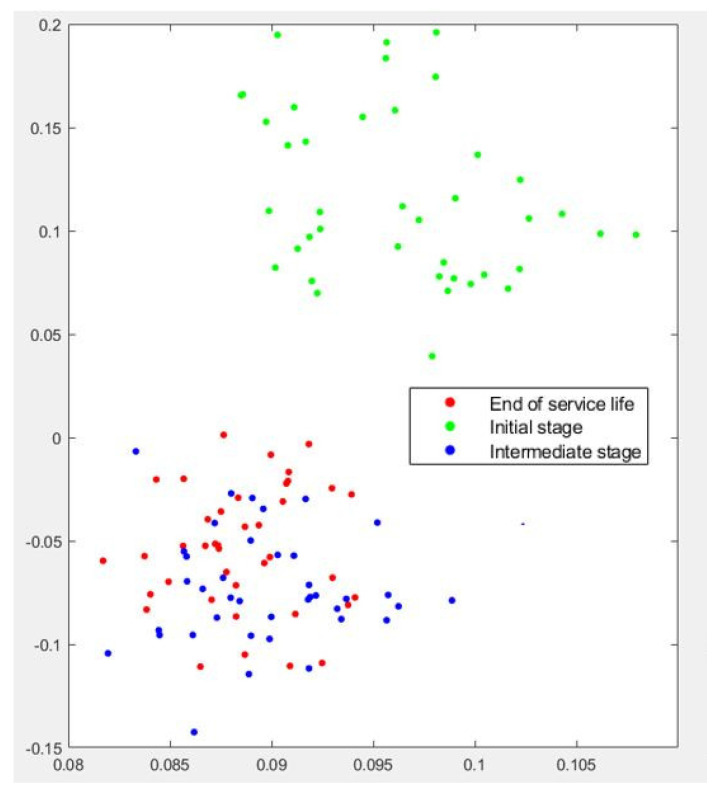
Graphical results of PCA analysis for Test No.3.

**Table 1 sensors-22-06911-t001:** Definition of the exerimental tests.

Conditions for the Grinding Tests			
TEST No. 1: b_s_ = 10 mm	a_e_ = 0.030 mm	v_w_ = 25,000 mm/min	v_s_ = 30 m/s
TEST No. 2: b_s_ = 10 mm	a_e_ = 0.010 mm	v_w_ = 15,000 mm/min	v_s_ = 30 m/s
TEST No. 3: b_s_ = 10 mm	a_e_ = 0.030 mm	v_w_ = 15,000 mm/min	v_s_ = 30 m/s
**Dressing conditions**			
	a_d_ = 0.030 mm	V_d_ = 130 mm/min	v_s_ = 30 m/s
**Coolant**			
5% water-based oil emulsion		Q = 8 L/min	Pressure = 1 bar

## Data Availability

Not applicable.

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
