# Peer review of "Deep Learning-Based Feature Extraction of Acoustic Emission Signals for Monitoring Wear of Grinding Wheels"

_sensors, 2022, doi:10.3390/s22186911_

Round 1
Reviewer 1 Report
The article is devoted to an interesting and practically important topic of monitoring wear of grinding wheels. However, there are a few comments.
1. Authors should double-check formula 1, because the dimensions of the left and right sides does not match. [P] = W = J/s, [ae] = mm, [vw] = mm/min, so P / (ae*vw) = 60*J/mm2, however [ec] = J/mm3.
2. Authors should add labels on axis figures 2, 3, 4.
3. The authors should expand Section 3 and explain how exactly the result of the FFT processing of the AE signal (1D value of high dimension about 300_000) goes to the ResNet18 input (2D value of the dimension 224*224), as well as what is the dimension of the ResNet18 output and how it gets to the input t -SNE and PCA for further dimension reduction.
4. It follows from Section 4 that the methods used are good at separating the signals related to the Initial stage (green dots), and much worse distinguish between the Intermediate stage (blue dots) and End of service Life (red dots) using clustering in 2 dimensions. Can these results be improved by clustering in 3 dimensions?
Reviewer 2 Report
The results of t-SNE and PCA analysis show the excellent efficiency of the pre-trained CNN for automated feature extraction from FFT plots. However, some minor issues need to be addressed before publication.
1) Section 3.2: What are the inputs and outputs of the CNN in this method? What are the features extracted by the neural network from the frequency domain FFT plots?
2) Section 4: It seems that no results of the DL-based feature extraction were shown.
3) PCA is linear while t-SNE is a nonlinear dimension reduction algorithm. Why can the efficiency of CNN be judged by the performance of these two unsupervised algorithms?
4) In practice, many different industrial grinding conditions exit. Considering of the generalization performance of the neural network, whether is this method suitable for other grinding conditions?
Reviewer 3 Report
The article is devoted to the processing of audio signals by machine learning methods. The topic of the article is relevant. The structure of the article does not correspond to that adopted in the MDPI for research articles (Introduction (including a review of analogues), Models and methods, Experiments, Discussion, Conclusions Nomenclature). The article is easy to read. The level of English is acceptable. The quality of the figures is acceptable. I recommend to the authors in Fig. 7-12 use geometrically different symbols instead of multicolored dots to represent clusters. The article cites 34 sources, not all of which are relevant. The list itself is sloppy.
The following remarks can be made on the material of the article:
1. The idea of ​​using convolutional neural networks to analyze audio signals is not new. The signal is fed to the network input in its original form, and band-pass filters are used as convolutions, the parameters of which the network selects in the learning process. For each filter, only 2 parameters are trained - high and low frequencies. Thus, on the one hand, we allow the algorithm to see raw data, and on the other hand, we teach the network to look at data in the context of only certain frequency ranges. An obvious breakthrough in this direction of using machine learning came in 2019 with the advent of the SincNet architecture. Both networks (CNN and SincNet) ultimately learn the same thing, namely the selection of certain frequencies of interest in the signal, however, SincNet has a certain trump card - information about how the filter looks like is added to the first layer of this network, while like a conventional CNN is forced to choose the best filter shape on its own. Thanks to this approach, SincNet has a number of advantages over standard convolutional networks, namely: 1. Fewer parameters (each SincNet convolution always depends only on 2 parameters - lower and upper frequencies); 2. Fewer operations to calculate the filter, since the convolution is symmetrical. We calculate the first half, mirror the second; 3. Fast convergence. This is a consequence of the first two points; 4. Good interpretability of the network - each convolution is a filter with clear boundaries. At the same time, the normalized sum of filters indicates the focus of attention on the main tone and formants. All these advantages are supported by comparisons of quality metrics, where SincNet shows better results than the classic DNN-MFCC, CNN-FBANK, CNN-RAW bundles. So, the SincNet architecture is not mentioned at all in the article! Why?
2. The ultimate goal of the authors is not classification, but clustering. This means that the working architecture must be trained in such a way that the vectors from the output of the second model (in fact, the embedder) are close in n-dimensional space if they belong to the same class and far away if they belong to different classes. For this, the MetricLearning technique is used in the learning process. An example code for such training can be found, for example, here: https://github.com/drova326/sincnet_metric_learning. The authors chose a long and untested path. Why?
3. To reduce the level of errors of the second kind, it is recommended to pretrain neural networks for analyzing audio signals on a dataset, in the markup of which we are 100% sure. I didn't notice that the authors use this technique. Why?
4. To estimate the proximity of the obtained vectors in the context of the clustering problem, the UMAP dimensionality reduction method is usually used. The authors were original here too. Why?
5. Despite the apparently good performance of the vectorizer, the clustering accuracy (judging by Fig. 7-12) is somewhat confusing. I think that the reason is that this accuracy is considered in the context of the batch, and random parts from the audio get into the batch, including those that cannot be used to clearly judge which cluster the signal fragment belongs to (for example, a short pause or extraneous noise ). To assess the quality of the model, a slightly different approach is used: the signal is divided into short sections, each section is classified, and after that the most frequently predicted label is assigned to the entire set.
Round 2
Reviewer 3 Report
First of all, I note that I like the calm and polite style of the authors.
I made the following recommendations for the basic version of the article:
1. The idea of using convolutional neural networks to analyze audio signals is not new. The signal is fed to the network input in its original form, and band-pass filters are used as convolutions, the parameters of which the network selects in the learning process. For each filter, only 2 parameters are trained - high and low frequencies. Thus, on the one hand, we allow the algorithm to see raw data, and on the other hand, we teach the network to look at data in the context of only certain frequency ranges. An obvious breakthrough in this direction of using machine learning came in 2019 with the advent of the SincNet architecture. Both networks (CNN and SincNet) ultimately learn the same thing, namely the selection of certain frequencies of interest in the signal, however, SincNet has a certain trump card - information about how the filter looks like is added to the first layer of this network, while like a conventional CNN is forced to choose the best filter shape on its own. Thanks to this approach. All these advantages are supported by comparisons of quality metrics, where SincNet shows better results than the classic DNN-MFCC, CNN-FBANK, CNN-RAW bundles. So, the SincNet architecture is not mentioned at all in the article! Why?
2. The ultimate goal of the authors is not classification, but clustering. This means that the working architecture must be trained in such a way that th vectors from the output of the second model (in fact, the embedder) are close in n-dimensional space if they belong to the same class and far away if they belong to different classes. For this, the MetricLearning technique is used in the learning process. An example code for such training can be found, for example, here: https://github.com/drova326/sincnet_metric_learning. The authors chose a long and untested path. Why?
3. To reduce the level of errors of the second kind, it is recommended to pretrain neural networks for analyzing audio signals on a dataset, in the markup of which we are 100% sure. I didn't notice that the authors use this technique. Why?
4. To estimate the proximity of the obtained vectors in the context of the clustering problem, the UMAP dimensionality reduction method is usually used. The authors were original here too. Why?
4. Despite the apparently good performance of the vectorizer, the clustering accuracy (judging by Fig. 7-12) is somewhat confusing. I think that the reason is that this accuracy is considered in the context of the batch, and random parts from the audio get into the batch, including those that cannot be used to clearly judge which cluster the signal fragment belongs to (for example, a short pause or extraneous noise ). To assess the quality of the model, a slightly different approach is used: the signal is divided into short sections, each section is classified, and after that the most frequently predicted label is assigned to the entire set.
The authors to some extent took into account all of them. To be honest, any text larger than a page can be improved endlessly ) But I will not insist on this. The article can be published. The weaknesses of the study will become the strengths of new articles by the authors. I wish the authors successful scientific research.